# Towards Deep Viticultural Representations: Joint Region and Grape Variety Embeddings

## Abstract

Characterizing the terroir of viticultural regions (areas growing grapes for wine) in a data-based manner globally is a major aim of viticultural research. An associated challenge is selecting fitting grape varieties for viticultural regions under a changing climate. Furthermore, for breeding and variety selection categorizing the over 1557 commercially grown grape varieties is crucial. To reduce the dimensionality of these problems linear dimensionality reduction techniques are currently used such as PCA, based on hand-picked climate features, which leaves out many other important terroir characteristics such as soil, micro-climate, aspect and slope to name a few. For classifying grape varieties the current classifications do not account for viticultural characteristics but rather their origin, which is not directly useful for practitioners. As it is difficult to globally obtain the exact conditions under which every grape-variety is grown, in this study a method is proposed that can approximate grape variety similarity by relating them to their spatial distribution. This is achieved through joining the Variational Autoencoders approach, which allows for a non-linear dimensionality reduction for region and grape varietal classification; it is expandable and also already shows promise for making data-based varietal selection recommendations. To our knowledge this is the first time that autoencoders are adapted to the purely categorical domain; we demonstrate balancing of the single VAE loss function for this application. Additionally, we present a methodology to balance the joining losses of joint VAE's using a Tree-structured Parzen Estimator (TPE) approach, instead of balancing them by hand. We demonstrate meaningful representations by applying them to classify regions and varieties and beating classic single-value decomposition as well as climate informed PCA on regional climate classification as well as variety classification tasks.

## 1 Introduction

### 1.1 Motivation

Terroir is a combination of characteristics that affect the quality of grapes grown for wine and consists of the soil, climate and other micro-climatic factors (Leeuwen & Darriet, 2016; Leeuwen et al., 2018). Viticulture (growing grapes for wine) relies on choosing appropriate grape varieties and best sites to maximize the quality of grapes and wine. In order to do this characterizing the terroir of viticultural regions in a data-based manner globally is a major aim of viticultural research such as Cardoso et al. and Puga et al. (2022). As the current climates are changing this is made more difficult. Well-studied climates in well known grape growing regions are shifting significantly already endangering grape and wine quality (Leeuwen & Darriet, 2016). New varieties may soon be needed, and recently even highly regulated regions such as Bordeaux have already approved new varieties to be used in its appellation. Additionally, general representations for the over 1557 commercially grown grape varieties are crucial for understanding where lesser known varieties could thrive. Currently, viticultural data scientists utilize quite simple linear dimensionality reduction techniques on hand-defined climate features. As terroir is a complex concept depending on the mutual effects of macro and micro-climate which are themselves influenced by geographical attributes such as slope and aspect and cultural aspects like the training system and various soil characteristics climate averages cannot encompass its true effect (Leeuwen et al., 2018). Even more so, for grape varieties

the focus is on heritage rather than viticultural characteristics (Bisson, 1999; 2009; Levadoux et al., 1948; Levadoux, 1956). Though heritage may indicate some similarity in historical use between grape varieties, it does not take into account current cultivation experience. The current globalized wine market has demonstrated that some previously local varieties thrive even more under new-world conditions. As such, predicting which varieties could have similar success in a region's future climate would be very useful to facilitate the climate-induced viticultural transition.

From a methodological standpoint there are some requirements for a solution to this problem. We frame the problem as a generative problem (i.e. grape variety recommendations are generated); so the latent space of the generative model provides the reduced dimensionality terroir-space and low dimensional variety similarity space for classification or clustering tasks. The latent space of the chosen model should furthermore be continuous. Latent space continuity is important to prevent non-sensible variety recommendations from 'white space' in the latent space.

We would furthermore like to align/mirror the human representations of wine experts in our embeddings. Otheguy et al. (2021) found that most experts focus on two main aspects when thinking about wine. These were identified to be "environment and the vineyard" (Otheguy et al., 2021), which we account for by utilizing the growing region of the grapes, and "the intrinsic aspects of wines"(Otheguy et al., 2021), which we incorporate through grape varieties. Both our choices of key variables represent abstractions of the concepts that they stand for. However, they allow for approximation of these concepts, as the environment a grape is grown in is linked directly to the region it is grown in and the wine style is made possible/limited by the grape-variety used. Uncertainty must be accounted for in the latent space as well, as no part of wine production is a completely controllable process. Rather than modelling a value for each variety and region the representation learner should model a multivariate distribution for this inherent variability.

## 1.2 RELATED WORK

Reducing dimensionality of complex problems has been a common problem in machine learning for a long time, and is often also referred to as embedding learning, especially when related to learning vectors from abstract data-types like words. The most prominent ML embeddings likely are word embeddings which started a revolution in Natural Language Processing (NLP), from shallow beginnings of Word2Vec (Mikolov et al., 2013) and GLOVE (Pennington et al., 2014) which condensed dictionaries with 1000s of entries to about 300 dimensions using the word-to-be-embedded's context words. Initial methods used were quite similar to shallow versions of autoencoders for the Word2Vec (Mikolov et al., 2013). Deep representations or embeddings of words and sentences eventually evolved into models such as BERT (Devlin et al., 2019), using the Transformer architecture. Though to this day in fields for which the vocabulary is quite task specific, custom word embeddings are still learned from relatively small corpi using models such as autoencoders (Bhardwaj et al., 2022).

To facilitate variety recommendation and learn variety and region embeddings, word embeddings techniques are not directly usable. This is due to the difficulty of finding a large corpus that adequately describes all grape varieties commercially produced and relates them to growing conditions. The most comprehensive book on commercial wine grape varieties is likely the book: "Wine grapes: a complete guide to 1,368 vine varieties, including their origins and flavours" by Robinson et al. (2013). Of course this book, although filled with expertise, cannot by itself provide a balanced corpus necessary to learn general grape variety embeddings. Even if other books and articles are added the corpus would remain unbalanced and favour important international grapes for embedding creation while embeddings for lesser-known and more locally important grapes would be significantly worse defined. Varietal and regional embeddings therefore should not be based on word embeddings but should be defined by other means. In this work we propose the use of a joint embedding between viticultural regions and grape varieties.

As demonstrated by Radford et al. (2021) for text and images and Cohen Kalafut et al. (2023); Radhakrishnan et al. (2023); Singh et al. (2023) for various bio-medical applications. Joining models 'cross-modaly' or between various types of input can lead to models and representations that have great multi-use-ability. Joint models are generally used for reducing various modalities of a common task can be reduced in one representation, which is an important property for being able to incorporate a variety of terroir characteristics. Additionally, the joint VAE approach further allows

imputing or encoding from one modality and decoding from the latent space for another modality (Cohen Kalafut et al., 2023), which is what is required for variety recommendation. Generally encoders are trained by utilizing co-occurrence of different data of the same sample such as images and their description (Radford et al., 2021). For the joint VAE approach for each modality a decoder is also used that re-constructs the modality from the joint latent space (Radhakrishnan et al., 2023), which is what allows for construction of one modality from the latent space of the other. Further, the KL-divergence component of the loss function commonly used with VAEs requires the model to learn a somewhat continuous embedding space that can be interpolated (Kingma & Welling, 2022). VAEs also model a multi-normal distribution rather than an absolute embedding vector; the embedding vector is sampled from the multi-normal distribution at every pass through the model (Kingma & Welling, 2022), which allows for the incorporation of uncertainty. For this reason, we choose a joint VAE approach for this paper.

Since for viticulture we do not necessarily have the specific location and associated soil, topography and micro-climate in order to join the latent spaces we use the growing regions as an approximation of these. In this work we use the distribution of grape varieties within a region as the basis for the regional and varietal embeddings based on the "Database of Regional, National and Global Wine-grape Bearing Areas by Variety" (Anderson & Nelgen, 2020). Our approach allows sampling the soil, micro-climate and geographical characteristics of a region to connect the terroir characteristics from aggregate statistics rather than fine-scale data. Hence the region embedding for now serves as the approximation of terroir, and can in the future be used to also join an arbitrary number of terroir characteristics for the regions where it is available. This approach sets the foundation of a global terroir embedding as we define region and variety similarity.

## 2 MATERIALS & METHODS

### 2.1 DATASET GENERATION

The main goal of the generated datasets is to assure proper correlation for the models to be able to use co-occurence of varieties and regions while also ensuring similarly strong representations for all wine regions and grape varieties regardless of popularity or total growing area. In order to achieve this we create two datasets from the regional bearing areas by variety of 2016, contained in the dataset by Anderson & Nelgen (2020), by a sampling technique described in more detail below. The created datasets will be referred to as the regions dataset ($RD$) and the variety datastet ($VD$). Let $R$ and $V$ be the sets of all regions and all grape varieties respectively and $A_{i,j}$ the bearing areas for each region $i$ and each variety $j$. The datasets are then constructed by iterating over all regions $\boldsymbol{r}_i$ and sampling a variety $\boldsymbol{v}_j$ at every epoch, weighted by the fraction of growing area it occupies in the region. This is described in following equation:

$$rd_i = \{\boldsymbol{r}_i, \boldsymbol{v}_j\} \text{ for } \boldsymbol{v}_j \in V \text{ such that } P(\boldsymbol{v}_j|\boldsymbol{r}_i) = \frac{A_{i,j}}{\sum A_{i,:}}$$

(1)

This is done similarly for varieties and $VD$, as shown below:

$$vd_i = \{\boldsymbol{r}_j, \boldsymbol{v}_i\} \text{ for } \boldsymbol{r}_j \in R \text{ such that } P(\boldsymbol{r}_j|\boldsymbol{v}_i) = \frac{A_{i,j}}{\sum A_{:,i}}$$

(2)

By iterating through each region for $RD$ and for each variety in $VD$ we ensure that per epoch every region and variety is shown to the model at least once. Further, by sampling the variety weighted by the proportion of its growing area for every region $r_i \in R$ to create $RD$, we generate a dataset that captures co-occurrence (as measured by the growing area). By generating $VD$ similarly and training representation learners in alteration on each dataset we are able to ensure that each region and each variety is represented somewhat in the dataset while not inhibiting co-occurrence to be the main driver for pairs of samples.

Every region $r_i$ consists of two features, first the region ($x1_i^r \in \{0, 1, ..., 594\}$), second the country ($x2_i^r \in \{0, 1, ..., 54\}$) that the region belongs to. The model receives each $r_i$ as two concatenated one-hot encoded vectors of a combined size of 650 input features. Each region $v_i$ consists of three features, first the variety ($x1_i^v \in \{0, 1, ..., 1556\}$), second the country of origin ($x2_i^v \in \{0, 1, ..., 54\}$) of the variety, and thirdly the colour of the grape variety ($x3_i^v \in \{0, 1, 2\}$). As a one-hot encoded vector it has 1650 input features.

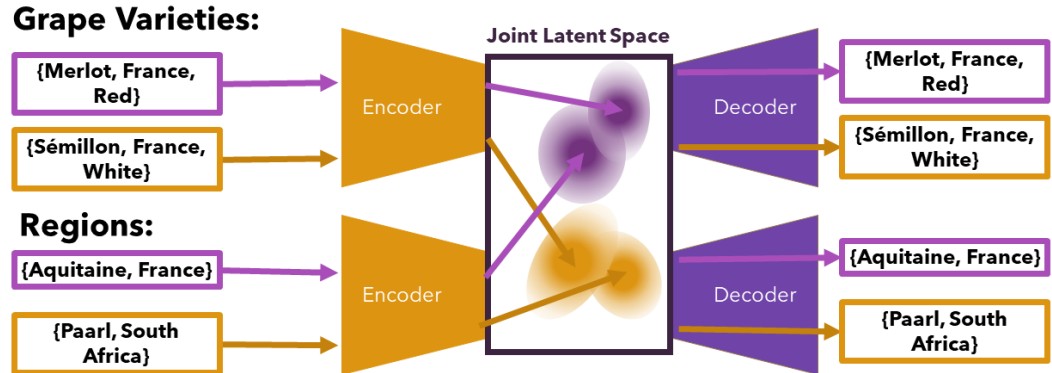

Figure 1: Illustration of the proposed solution, the joint VAE embedding learners.

## 2.2 REPRESENTATION LEARNERS

In order to create a desirable latent space to jointly represent regions and grape varieties we join two VAEs as shown in Figure 1. The VAEs used consist each of an encoder, latent-space projection heads, a decoder and classification projection heads. Encoders have 3 hidden layers of 1024 hidden units with sigmoid activation; we apply layer-norm after each hidden layer. The encoder is followed by two projection heads $\mu$ and $log\_var$ of the same dimension as the latent space. These linear projection heads model the multivariate mean and log variance of the input-representation in the latent space. Decoders are designed like the encoders with three layers of size 1024 and sigmoid activation as well as layer-norms after each hidden layer. Each output feature has its own linear classification projection head, i.e. two for the regions model and three for the variety model. We find the activation function through Tree-Structured Parsen Estimator (TPE) optimization from {GeLU, LeakyReLU, Sigmoid, Softsign} as well as the batch size of 32[1] from {32, 64, 128, 256, 512, 1024} and the learning rate of each model (region-model: $1.11\text{x}10^{-4}$, variety-model: $3.86\text{x}10^{-5}$). Let $decoder'$ include the decoder and classification heads for each model, the overall model configuration is then (Kingma & Welling, 2022):

$$\boldsymbol{z}_r = \mu_r(encoder_r(\boldsymbol{r})) + \epsilon_r * \exp\left(0.5 * log\_var_r(encoder_r(\boldsymbol{r}))\right) \tag{3}$$

$$\boldsymbol{r}' = regions\_model(\boldsymbol{r}) = decoder'_r(\boldsymbol{z}_r) \tag{4}$$

$$\boldsymbol{z}_v = \mu_v(encoder_v(\boldsymbol{v})) + \epsilon_v * \exp\left(0.5 * log\_var_v(encoder_v(\boldsymbol{v}))\right) \tag{5}$$

$$\boldsymbol{v}' = variety\_model(\boldsymbol{v}) = decoder'_v(\boldsymbol{z}_v) \tag{6}$$

for $\epsilon_r, \epsilon_v \sim \mathcal{N}(\boldsymbol{0}_h, \boldsymbol{I}_h)$, where $h$ is the embedding dimension and $\boldsymbol{r}'$ and $\boldsymbol{v}'$ are reconstructed versions of $\boldsymbol{r}$ and $\boldsymbol{v}$ respectively.

## 2.3 LOSS FUNCTIONS

The standard loss function for VAEs consists of the reconstruction loss term ($\mathcal{RL}(\boldsymbol{r}', \boldsymbol{r})$ or $\mathcal{RL}(\boldsymbol{v}', \boldsymbol{v})$) and a Kullback-Leibler (KL) divergence term ($\mathcal{KL}(\boldsymbol{\mu}', \mathbf{log\_var}')$, where $\boldsymbol{\mu}'$ and $\mathbf{log\_var}'$ are the outputs of the $\mu$ and $log\_var$ projection heads)[2]. KL-divergence is used here to force the model to define a more continuous latent space by rewarding non-zero variance and low means thereby ensuring a tighter latent space and distributed embeddings/representations for the same inputs. We found the need to balance the reconstruction and KL-divergence components of the losses for our purely categorical application area. In experiments without scaling the reconstruction loss we find that it is not reduced. Instead of the usual mean-square-error (MSE) loss used

---

[1]For training correlation-loss based models, mentioned later, the batch size is 512 as this loss needs to approximate the overall dataset.

[2]We use the form as described in eq. 10 of (Kingma & Welling, 2022)

for reconstruction (Kingma & Welling, 2022), we use the sum of cross-entropy loss for each reconstructed feature[3]. We found the MSE loss unsuitable for our application with large one-hot-encoded vectors, the cross-entropy loss is more appropriate, but requires balancing. We use $10^3$ to balance the reconstruction loss for our paper, and we refer to the reconstruction loss coefficient as $c_{rl}$. This is because in various experiments not shown here we found that with $c_{rl} < 10^3$ only the KL-term is reduced by the model.

To join the region and variety model we add a joining loss term ($\mathcal{JL}(z, z')$) as is done in previous literature on multi-modal autoencoders Radhakrishnan et al. (2023); Li et al. (2019). There are three types of losses commonly used as a joining loss, correlation-based losses, similarity-based losses and distance-based losses (Li et al., 2019). We test one of each category in order to evaluate which trains a model to fulfill the requirements outlined above. The overall loss then has the general form:

$$\mathcal{L}(y, y', \mu', \log\_var', z, z') = c_{rl} * \mathcal{RL}(y', y) + \mathcal{KL}(\mu', \log\_var') + c_{jl} * \mathcal{JL}(z, z')$$
(7)

where $y$ and $y'$ are either $r$ and $r'$ or $v$ and $v'$ depending on the model.

For correlation loss we utilize the canonical correlation analysis (CCA) based loss as described in Wang et al. (2015), here denoted as $\mathcal{C}(z, z')$. The correlation loss we use is inspired by more "classical", statistical methods that are used for multi-view alignment or multi-modal fusion prior to deep learning based methods (Wang et al., 2015). Being based in CCA it requires a good representation of the overall distribution for the whole dataset, therefore we use a different, larger batch size (512) for the CCA loss when training (Wang et al., 2015).

The similarity loss is implemented based on Radford et al. (2021) rather than only using the cosine similarity as described in Radhakrishnan et al. (2023). The similarity loss uses a contrastive setup where basically the cross-entropy of the dot-product between the latent variables of the region and variety batch is minimized (please see Radford et al. (2021) for the formulation). We omit the temperature term described by Radford et al. (2021) as we scale the loss to balance it with the other two loss terms in the next step.

The distance based loss is defined to ensure embedding in the same space while avoiding model collapse, which could otherwise occur with such a loss according to (Mialon et al., 2022). We measure distance in this case using the L2-norm. We treat the batch of embeddings from the regions model and the batch from the variety model as one batch (by concatenation) and perform feature-wise standardization before calculating the L2-norm on the difference between the jointly standardized batches.

As previously mentioned we use the weights $c_{rl}$ and $c_{jl}$ for the joining loss to balance the loss function to optimize all three component losses. Previously, in the literature such as Radhakrishnan et al. (2023), joining losses are balanced by hand-picking. Here, we instead find joining weights by TPE-optimization on both models simultaneously, to make this approach more rigorous and to avoid any bias for loss-evaluation. The criterion we minimize is based on minimizing the difference between average join and reconstruction loss improvement over 5 epochs of training. This is done such that during training both are optimized to the same degree. The optimization criterion is defined as follows:

$$\mathcal{L} = |(c_{jl,v} * \frac{jl_{5,v} - jl_{0,v}}{jl_{0,v}} + c_{jl,r} * \frac{jl_{5,r} - jl_{0,r}}{jl_{0,r}}) - ((rl_{5,v} - rl_{0,v}) + (rl_{5,r} - rl_{0,r}))|$$
(8)

where $jl_{e,y}$ and $rl_{e,y}$ are the joining and reconstruction loss respectively at epoch $e$ and modality $y$. The exact weights found and used are given in table 4 in appendix A.1. Readers interested in the computational requirements of our approach are referred to table 5 in the appendix A.1. For computational complexity we expect our approach to be comparable to published literature cited here such as Cohen Kalafut et al. (2023) and Radhakrishnan et al. (2023).

---

[3]This means: variety, country-of-origin, colour for the variety model. And, region and country for the region model.

## 2.4 TRAINING

In total 18 models are trained[4] in 9 configurations to estimate performance dependence on embedding dimension for which we use $\{4,12,36\}$ and dependence on joining loss function for which we use $C(\boldsymbol{z}_r, \boldsymbol{z}_v)$, $S(\boldsymbol{z}_r, \boldsymbol{z}_v)$ and $D(\boldsymbol{z}_r, \boldsymbol{z}_v)$. We use one AdamW optimizer for each model as used in Radford et al. (2021), with a weight decay of $10^{-4}$ according to Wang et al. (2015) and Lopez-Paz et al. (2014), and a learning rate as described in 2.2. All models are trained for 500 epochs, alternating between the $RD$ and $VD$ every 5 epochs. For model abbreviations "r" or "v" indicate whether a regions or variety model is used, the second letter "c", "d" and "s" indicate the joining loss used (correlation loss, distance loss, and similarity loss respectively). The last number in the abbreviation indicates the dimensionality or size of the latent space; this may either be 4, 12 and 36.

## 3 RESULTS & DISCUSSION

To understand whether representation learners have succeeded one must evaluate the amount of useful information they manage to incorporate. Here, information content of the latent-space is tested by applying each model to a potentially useful downstream task. First, region embeddings are evaluated.

## 3.1 CLIMATE CLASSIFICATION

The downstream task used here is to predict how the European Union would classify a specific European wine region, using a KNN classifier similar to approaches currently used in viticultural ML (Puga et al., 2022). We construct a test-dataset[5] out of the European Union (EU) classification system legislation document (European-Comission, 2008). As our regions are defined by the growing area by variety database(Anderson & Nelgen, 2020), our resolution is limited to that of the database. This does not allow us to always match the regions clearly to one of the 6 defined classes, A, B, C I, C II, C III (a), C III (b). Therefore, we keep only the major 3 classes A, B and C and also omit any region which with our resolution is split between the major classifications, to avoid confusion. The dataset includes 150 regions and testing error is calculated on a 40% test set (60 regions). We choose a large test-set to highlight the utility of our representations for enhancing predictions on small or incomplete datasets, and overcoming data-scarcity. The wine classification of the EU is of interest as it defines these classes by major terroir groups, based on climate and elevation. One of the main goals of this research is to eventually be able to match grape varieties to an appropriate terroir, so representing similar regions and their terroir similarly is a pre-requisite for this. The classification of regions by the EU is done to prescribe which policies govern them (European-Comission, 2008), so inadvertently, models that perform well on this test may also contain some intrinsic representation of the practices of regions.

Mean accuracy over 50 runs is used as the evaluation metric from 50 pseudo random runs to account for the stochastic nature of model design processes, and prevent bias in reporting of results. For every run a random validation set is created that is used in determining the ideal number of neighbours, and similarly, every run also has a different randomly selected test-set. Our baselines are evaluated likewise. The first baseline uses a truncated Single-Value Decomposition (SVD) of the transpose of the region-wise scaled version of the co-occurence matrix $A$. The second baseline is a PCA-decomposition. The PCA-baseline is based on the paper by Puga et al. (2022), which is the first and so far only global wine region classification paper. Puga et al. (2022) use the averages of 16 climate indexes of a "representative location", a city, in a wine region, and classify them using PCA-reduction to three components and subsequent K-means clustering. As their study is based on a similar database to ours many regions are equivalent, however, overall their regions are more finely defined. We create the equivalency by determining which of their regions has the greatest growing area in one of our regions, as their regions always fall completely into only one of our wine regions. We perform PCA decomposition on the 16 climate indexes they provide for all equivalent regions (595 regions). The explained variance ratio of the PCA baselines is 99% for 4 components,

---

[4]Associated codes as well as pre-trained models will be made available upon publication of this paper on GitHub.

[5]Available in the supplemental materials.

Table 1: EU-Region classification using region embedding and KNN (50 run mean accuracy). The first row is based on KNN classification using 1 sample and the second row using 10 samples from each encoder per region (on the first and second row respectively). Values are compared with a truncated-SVD baseline constructed on the scaled transpose of $A$ in the first row and a PCA analysis based on 16 climate averages from Puga et al. (2022) in the second row.

| b_36 | b_12 | b_4 | _c36 | **_s36** | _d36 | _c12 | **_s12** | _d12 | _c4 | **_s4** | _d4 |
|---|---|---|---|---|---|---|---|---|---|---|---|
| .747 | .734 | .701 | .850 | **.887** | .794 | .830 | **.899** | .858 | .804 | **.909** | .857 |
| 10x&PCA: | .753 | .731 | .866 | **.895** | .840 | .845 | **.904** | .867 | .818 | **.912** | .863 |

and 100% for 12 components; the original paper by Puga et al. (2022) used three components. The results of this experiment are shown in Table 1.

Both methods based on more classical decomposition approaches are outperformed by our deep-learning approach consistently. PCA based on climate data has a slightly better performance than the truncated SVD approach that had no access to climate data. However, the joint VAE approach introduced here outperforms these both completely without needing any climate data, which the EU classifications are based on. Therefore, we conclude that utilizing joint VAE's to learn complex relationships from relatively simple multi-modal co-occurrence datasets has the potential to extract meaningful physical information and to match similar terroirs to each other. Further, we observe trends towards higher performance the lower the dimensionality of the input is for our models, though for the baselines the trend is seemingly the opposite. With a classifier such as KNN it is expected to perform better as the dimensionality sinks as it is very susceptible to the curse of dimensionality and any unnecessary dimensions will significantly lower the classification accuracy (Keogh & Mueen, 2017). The fact that baselines are not able to take advantage of this phenomenon as the VAE's may relate to SVD and PCA both truncating components whereas the VAE's simply define a new latent-space that fulfills the same requirements as the larger latent-space.

We also run the experiment with 10 samples per region which increases performance for all our models. Interestingly, though the similarity-based model performs the best at all dimensions and for 1 and 10 samples per region, increasing the number of samples per region has a smaller effect on s-model-feature performance than it has for other models. This, as will be covered in more detail later in section 3.3, relates to the latent-space continuity of the similarity-loss based models.

## 3.2 VARIETY CLASSIFICATION

To determine the utility of features of the variety models we train a linear-probe to classify grape varieties to eco-geo-groups (Levadoux et al., 1948; Levadoux, 1956; Bisson, 1999; 2009), as presented by Robinson et al. (2013). The eco-geogroups represent the best baseline classification system we could find which incorporate some aspects of grape suitability for specific regions. The eco-geogroups, though primarily defined by heritage, are also linked to the areas of France that the grapes in that family originated from(Robinson et al., 2013). They therefore have a geographic, environmental, or by our definition, a terroir association. Unfortunately this dataset is very limited and some of the few varieties it contains are no longer commercially grown. Only 102 grape varieties could be assigned to 1 of 13 groups, making for a remarkably small data-set for ML tasks. Groups are quite unbalanced as well, even after removing groups with less than 5 member grape varieties, though it is the best available to our knowledge.[6] We utilize a baseline of truncated SVD features extracted from a variety-wise normalized version of the co-occurrence matrix $A$.

The results are shown in Table 2. We use average class-balanced accuracy over 10 runs as the evaluation metric, again accounting for varying test-set scenarios. Due to the limitations discussed above we utilize 1000 samples per variety to augment the dataset. The ability for such an augmentation is a valuable feature of the VAE approach and is unique, especially since all input variables are discrete, one-hot-encoded vectors. Of the 9 models used, 6 outperform their respective baselines. This is

---

[6]This test dataset is also available in the supplemental materials.

Table 2: Eco-georegion grape variety classification using variety embedding and linear probes, showing averages and standard deviations from 10 runs (1000 samples per variety), compared with a truncated-SVD baseline.

| b_36 | b_12 | b_4 | _c36 | **_s36** | _d36 | _c12 | **_s12** | _d12 | _c4 | _s4 | **_d4** |
|------|------|-----|------|----------|------|------|----------|------|-----|-----|---------|
| .238 | .138 | .031 | .151 | **.280** | .155 | .131 | **.274** | .168 | .104 | .102 | **.137** |

quite a positive result, though it is less impressive than the region-classification result. Again, the s36 and s12 models stand out but d12 and d4 also show good performance. Additionally, again in the smallest latent space of size 4 the VAE models seem to have an especially high advantage over classic methods. Combined with the results from section 3.1, we demonstrate that the representation learners shown here have managed to learn useful information that is useful for related ML tasks.

## 3.3 QUALITATIVE ANALYSIS

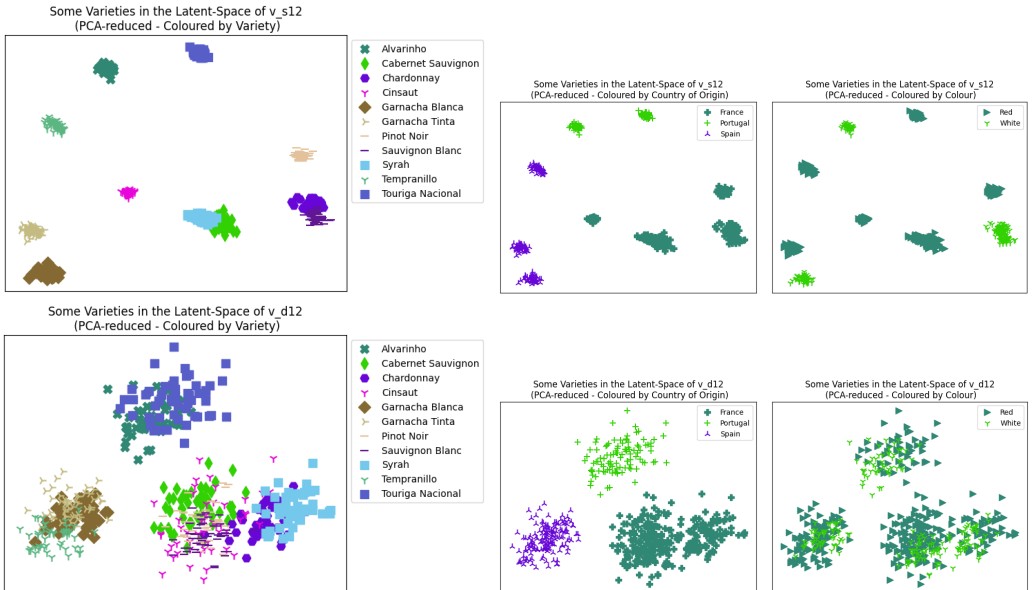

Figure 2: Views of a few French, Spanish and Portugese varieties in the latent space of v_s12 and v_d12. Showing 50 repetitions per variety. (Top: From left to right: 1. v_s12 latent space labelled by variety, 2. & 3. same latent space labelled by country and grape colour respectively. Bottom: same order but with v_d12 latent space.)

In this section we analyze the goodness of the learned representation qualitatively from tSNE and PCA reduced views of the latent spaces of the best performing model from the experiments above, r/v_s12. We additionally compare this with views of r/v_d12 which is a well-performing model for most tasks above and is able to give us insight into the different configurations of latent spaces that can be achieved using different loss-functions.

In Figure 2 it can be seen that both models more or less meaningfully assemble a latent space for various French, Portuguese and Spanish varieties shown. All countries of origin are easily separated, these are correlated with the heat-tolerance of the various varieties, being highest for the Spanish varieties and lowest for the French varieties. In the latent space of v_s12 even the fine details are meaningful. Not only are varieties separated by country of origin but also the arrangement of the varieties is meaningful. For example, Pinot Noir and Chardonnay and Sauvignon Blanc are well known to thrive in cooler wine-regions, being popular in Champagne, Germany and New Zealand. On the other hand Cabernet Sauvignon and Syrah are often grown in comparatively hot climates, so is Cinsaut, such as Southern France (i.e. near Spain), in the middle-east (Cinsaut), or Australia.

Cabernet Sauvignon is grown in nearly every warm to hot region in the world. The latent space of v_s12 keeps to these trends, by locating more heat-tolerant grapes near Spain and Portugal and keeping cooler-climate grapes further away. Even Alvarinho, which is a Portugese grape that is also grown frequently in Northern Spain is kept closer to Spain while Tinta Roriz which is more endemic to Portugal is further away. The v_s12 model even keeps grapes of the same colour near each other while maintaining these trends. This can be seen by the Chardonnay (white), Sauvignon Blanc (white), Pinot Noir (red) clusters. In conclusion, the latent-space of v_s12 is full of meaning, explaining it's high performance on above tasks. Unfortunately, the distance-loss based model clearly has much more overlap, especially in the details, for example between red and white grape varieties. For the v_d12 model it is rather difficult to determine, at least in the current PCA reduction view, exactly how meaningful the details of the latent space are.

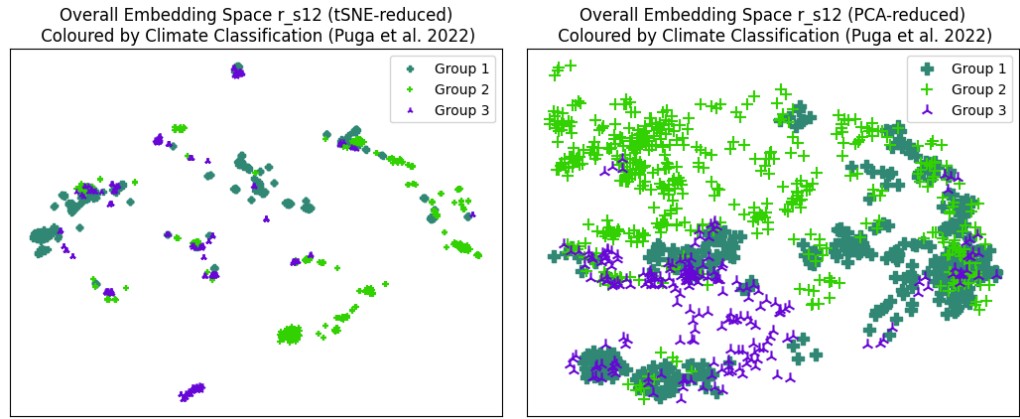

Figure 3: Views of the entire latent-space of the region model, showing three samples per region. This compares the views obtained from PCA and tSNE analysis.

Figure 3 shows the latent space of r_s12 reduced by both tSNE and PCA. The figure illustrates the entire latent space such that there are three samples per region for all regions. It shows It is observed that the tSNE reduction leads to a highly disconnected latent-space view whereas the PCA reduction maintains a larger amount of connectivity between regions. In fact, in additional experiments, shown in the appendix A.1 the continuity and overlap of the models are analyzed more rigorously using K-means clustering analysis. The analysis confirms the disconnection between clusters for all of the similarity-loss models, compared to their competitors. This is explainable by the contrastive nature of the loss. Contrastive losses "push away" all representations that are not of the same sample(Radford et al., 2021), whereas other losses, especially the distance loss only try to minimize the distance between true samples (Li et al., 2019).

### 3.4 RECOMMENDATIONS

As pointed out earlier, the end-goal of the present work is to eventually recommend grape-varieties in a data-based manner to growers for their specific location and environmental terroir characteristics. Therefore, an important test is the ability of the current setup to be used for variety recommendation. This may be done through cross-modal inference; this simultaneously examines the quality of the "joining" of the region and variety models. We use the normalized output logits of the variety model decoder as proportional to the likelihood of variety fit to a region. The input to this setup is a latent representation of the region of interest. In addition to the variety recommendation setup (region → variety) the region recommendation/imputation (variety → region) is also investigated in this test to examine the latent-space alignment of the current setup. Modality reconstruction, inter-modal imputation or cross-modal inference is a common task for joint VAE's such as shown by Cohen Kalafut et al. (2023).

To be useful for terroir/variety matching in the future, ideally the model could provide recommendations for varieties not grown in regions currently or for regions which do not yet contain the grape variety. Therefore, we cannot directly use scaled areas ($A_{i,:}$ or $A_{:,j}$) as labels. Instead we use the

Table 3: Region and variety recommendation performance with score means over all varieties for v→r and over all regions for r→v are shown.

| r → v | rec | p | v → r | rec | p |
|---|---|---|---|---|---|
| _c36 | .530 | .326 | _c36 | .531 | .396 |
| _s36 | .582 | .212 | _s36 | .660 | .216 |
| **_d36** | .572 | .169 | **_d36** | .754 | .146 |
| _c12 | .502 | .458 | _c12 | .510 | .475 |
| **_s12** | .611 | .119 | **_s12** | .694 | .200 |
| **_d12** | .625 | .151 | _d12 | .673 | .230 |
| _c4 | .522 | .380 | _c4 | .451 | .549 |
| _s4 | .568 | .235 | **_s4** | .693 | .216 |
| _d4 | .532 | .309 | _d4 | .636 | .301 |

Spearman p-value of positive correlation (Spearman r was previously used by Wang et al. (2015)) between the actual scaled growing area ranking compared to the output activation/logits based ranking to evaluate its performance. This way, as long as varieties actually grown are ranked more highly than the average grape variety, we can assume the model to be making somewhat reasonable recommendations. We additionally show the recall score, defining a positive ranking to be above average output activation, and corresponding true positives are varieties with non-zero cultivation area. The mean recall and Spearman p-value over all input regions or all input varieties is shown. The output activations for each input are determined from averaging 10 inferences. See Table 3 for the results of this test.

The v/r_s4, v/r_s12, v/r_d12, and v/r_d36 models perform the best. A positive correlation between the model and growing-area rankings is generally $< .5$, though the correlation is never significant. The best p-values indicate probabilities of 78-88% of positive correlation, indicating some potential of this type of set-up for predicting suitability scores of grape-varieties for particular regions in a data-based manner, though not in it's current form. As this is the first work where all commercial varieties can be ranked by their suitability to most viticulturaly important regions, this serves as a new baseline for future work. Recall data mirrors this result. Other methods to optimize models for imputing such as drop-out based connection as shown by Radhakrishnan et al. (2023), should be investigated to make these models useful past their internal representations and for data-based, terroir-specific grape variety recommendation. Tying further modalities in such a set-up such as climate variables to the regions may assist in ranking grape varieties to certain current or future climates, which needs to be investigated in follow-up work.

## 4 CONCLUSION

This paper represents the first step towards self-supervised deep-learning-based representations for viticultural regions and grape varieties as well as the potential application of the same methodology to generate grape variety suitability rankings to certain agro-ecological environments. Properties of the latent space were investigated, and it was found that similarity based losses produce the most useful representations for downstream tasks. The research demonstrated useful properties of the latent space outperforming both a SVD and climate-based PCA baseline on climate classification tasks without using any climate data itself. Similarly, the presented approach outperforms SVD approaches on varietal classification. The performance difference is more apparent for low-projection-dimensions meaning that VAE's are able to better use few features to encode data than classic decomposition methods. This work is additionally a first step to data-based varietal selection for specific environments, though in it's current form it cannot be directly applied in practice. However, in this work the initial baseline is set. We believe this work will assist in addressing climate uncertainty and allow for more informed long-term planning in the wine industry.

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

## A  APPENDIX

### A.1  MODEL/LOSS PARAMETER DETAILS

The losses for all models are balanced using $c_{rl}$ and $c_{jl}$. The values of these for each model are shown in Table 4.

### A.2  COMPUTATIONAL REQUIREMENTS

Computation times for training each model pair for 500 epochs are given in the Table 5. These times are based on training the model on a Intel (R) Core (TM) i7-10610U CPU @ 1.80 GHz x 8.

Table 4: Balancing weights for the joining losses of the form: $\mathcal{L} = c_{rl} * \mathcal{RL} + \mathcal{KL} + c_{jl} * \mathcal{JL}$; $c_{rl}$ is handpicked after experimentation and $c_{jl}$ is found by Tree-structured Parzen Estimator (TPE) optimization.

| **MODEL** | $c_{rl}$ | $c_{jl}$ |
|---|---|---|
| r_cca-loss | $10^3$ | $1.0 * 10^4$ |
| r_similarity-loss | $10^3$ | $1.0 * 10^0$ |
| r_distance-loss | $10^3$ | $1.1 * 10^0$ |
| v_cca-loss | $10^3$ | $1.5 * 10^4$ |
| v_similarity-loss | $10^3$ | $1.0 * 10^0$ |
| v_distance-loss | $10^3$ | $4.4 * 10^0$ |

Table 5: Training time of each model on CPU, training for 500 epochs.

| Model | training time (min:sec) |
|---|---|
| _c36 | 14:29 |
| _s36 | 14:58 |
| _d36 | 14:52 |
| _c12 | 14:45 |
| _s12 | 15:04 |
| _d12 | 14:50 |
| _c4 | 14:58 |
| _s4 | 14:50 |
| _d4 | 14:21 |

### A.2.1 COMPARING STRUCTURE WITH HUMAN EXPERT REPRESENTATIONS

We expect the latent space of the models to be structured similar to expert representations of regions and varieties. Therefore regions that produce similar wine, such as Australian wine regions should be closer to each other to the point of overlapping. Sensory clustering of their wines by experts creates very non-homogeneous clusters, in other words they are easily confused (Souza Gonzaga et al., 2020). The "confusability" property between similar regions is one we would want our latent space to have. However, we expect the difference between more distinct regions to be fairly clear, such as between Bordeaux and most Australian wine regions (Souza Gonzaga et al., 2020).

We test continuity of the embedding space using the silhouette-score metric after K-means clustering[7] (see Table 6 & Table 7). A silhouette-score of 0 is the target, to indicate that our clusters are close together, which we also infer to indicate latent-space continuity. The silhouette-score has been used to evaluate the clarity of word embeddings but the aim in this example was to reach a score near 1 as separation was desired (Bhardwaj et al., 2022). Generally similiarity-loss (s) based models performed the worst in this metric, at nearly every dimension size and nearly every clustering label. Models using Correlation (c) and distance-based (d) losses at dimensions of 36 and 12 consistently outperform other models in this metric.

The homogeneity score (Rosenberg & Hirschberg, 2007), is also shown in Table 6 and 7 and is used as a metric to determine whether and how consistent the representations are. Here a higher score is desired, though it is difficult to define an ideal score as explained above a certain degree of overlap should exist between the labels such that potentially similar regions or varieties share part of their possible representations. This is important to mimic the human representations of wines (Souza Gonzaga et al., 2020), in other words, we want to allow for a degree of "confusability". The degree of the homogeneity wanted is a future area of research as it may depend on the final application of the representations. In general, it is noted that the s-models have a higher degree of homogeneity in the K-means clusters labelled by region and variety than the c- or d-models. These

---

[7]The k in k-means is adapted to whichever label we are clustering to, we set it to the number of total labels so if labelled by country we use 55.

Table 6: Clustering on regions (R) and country (CO) (50 samples per region), evaluated by homogeneity score (HOM) and silhouette score (SIL).

| MODEL | R:HOM | R:SIL | CO:HOM | CO:SIL |
|---|---|---|---|---|
| **r_c36** | 0.599 | -0.006 | 0.944 | 0.118 |
| r_s36 | 0.889 | 0.131 | 0.835 | 0.326 |
| **r_d36** | 0.781 | 0.062 | 0.955 | 0.099 |
| **r_c12** | 0.595 | 0.002 | 0.951 | 0.247 |
| r_s12 | 0.883 | 0.167 | 0.876 | 0.403 |
| **r_d12** | 0.704 | 0.054 | 0.963 | 0.203 |
| r_c4 | 0.558 | 0.088 | 0.920 | 0.304 |
| r_s4 | 0.857 | 0.206 | 0.879 | 0.464 |
| r_d4 | 0.611 | 0.109 | 0.920 | 0.308 |

Table 7: Clustering on variety (V), country of origin (COO), colour (C) (50 samples per variety), evaluated by homogeneity score (HOM) and silhouette score (SIL).

| MODEL | V:HOM | V:SIL | COO:HOM | COO:SIL | C:HOM | C:SIL |
|---|---|---|---|---|---|---|
| **v_c36** | 0.596 | -0.012 | 0.958 | 0.095 | 0.004 | 0.066 |
| v_s36 | 0.849 | 0.051 | 0.782 | 0.340 | 0.006 | 0.206 |
| **v_d36** | 0.587 | -0.015 | 0.956 | 0.044 | 0.010 | 0.041 |
| **v_c12** | 0.584 | 0.009 | 0.936 | 0.260 | 0.651 | 0.171 |
| v_s12 | 0.857 | 0.081 | 0.830 | 0.444 | 0.009 | 0.216 |
| **v_d12** | 0.627 | 0.028 | 0.931 | 0.167 | 0.009 | 0.124 |
| v_c4 | 0.568 | 0.063 | 0.898 | 0.556 | 0.759 | 0.243 |
| v_s4 | 0.811 | 0.117 | 0.822 | 0.443 | 0.004 | 0.290 |
| v_d4 | 0.622 | 0.058 | 0.899 | 0.307 | 0.002 | 0.267 |

characteristics of s-models suggest that their latent-space is more disconnected than the latent space of c- and d-models. This is confirmed by PCA-reduced plots of a subset of clusters, see A.3.

## A.3    RAW LATENT VIEWS

Shown here are some PCA reduced visualizations of the embedding space of various selected models, compared with tSNE reduced views of the same latent-spaces.

Some Varieties in the Latent-Space of v_s12
(PCA-reduced - Coloured by Variety)

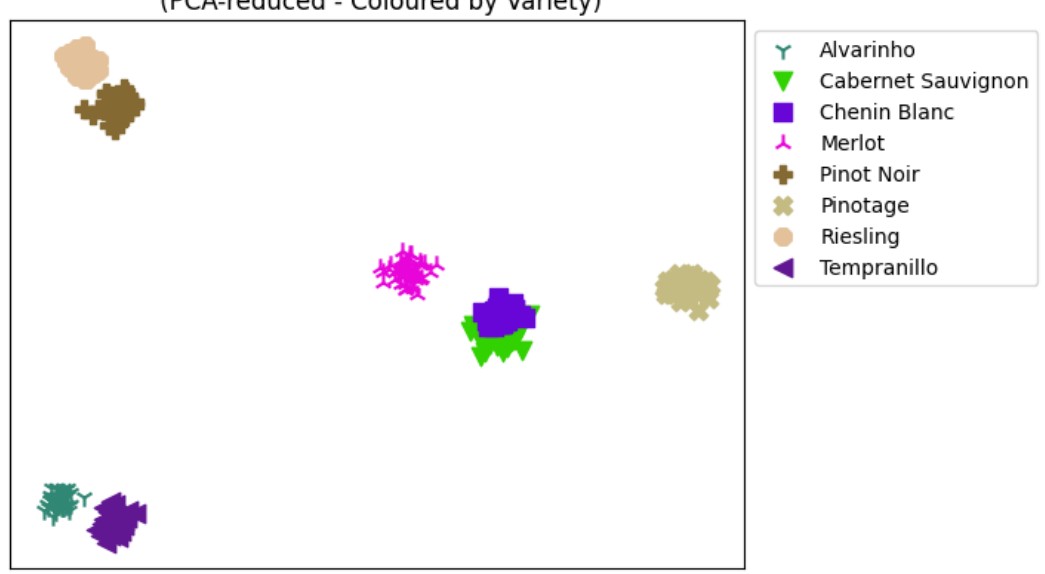

Some Varieties in the Latent-Space of v_d12
(PCA-reduced - Coloured by Variety)

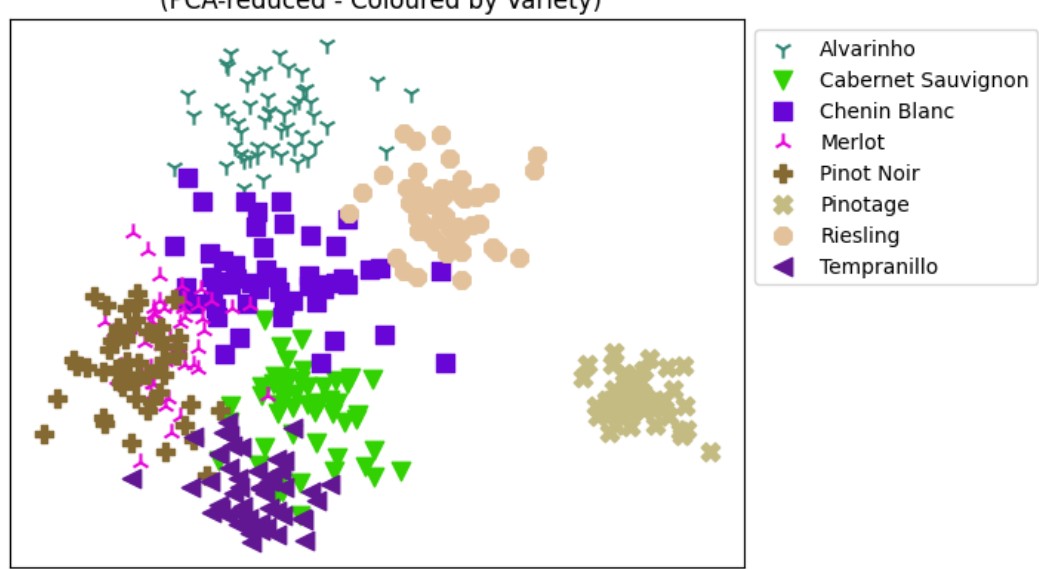

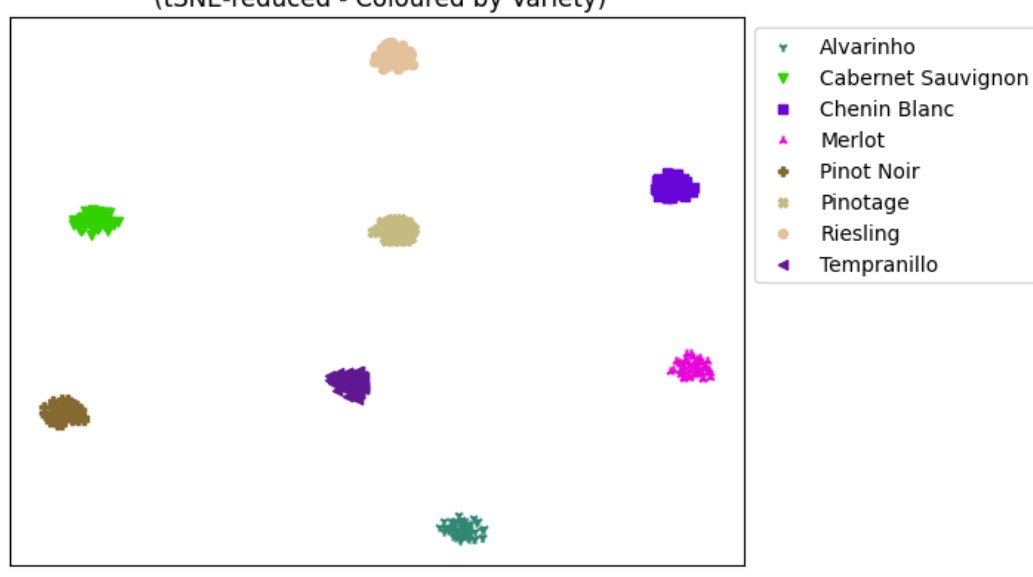

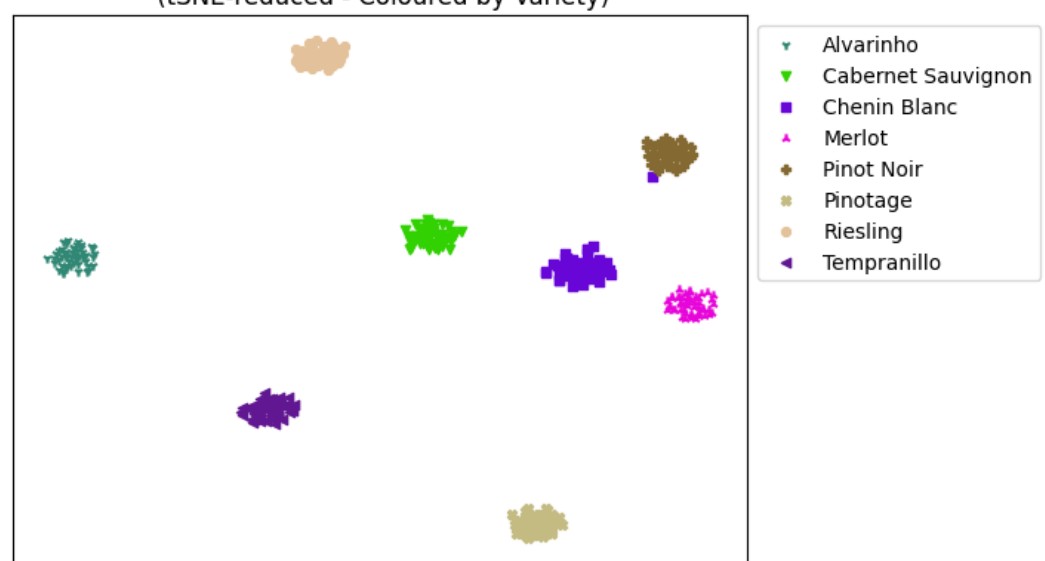

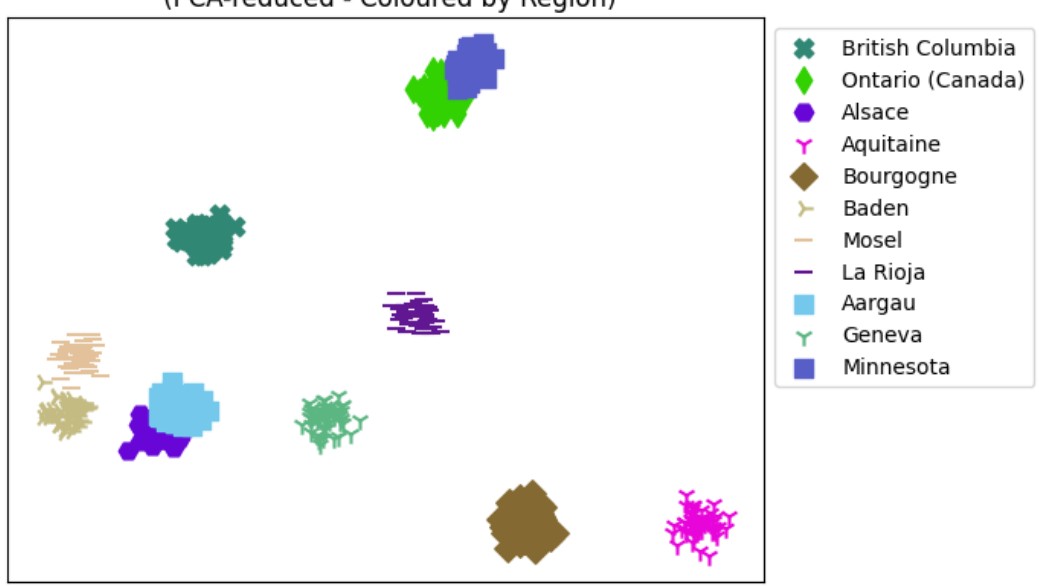

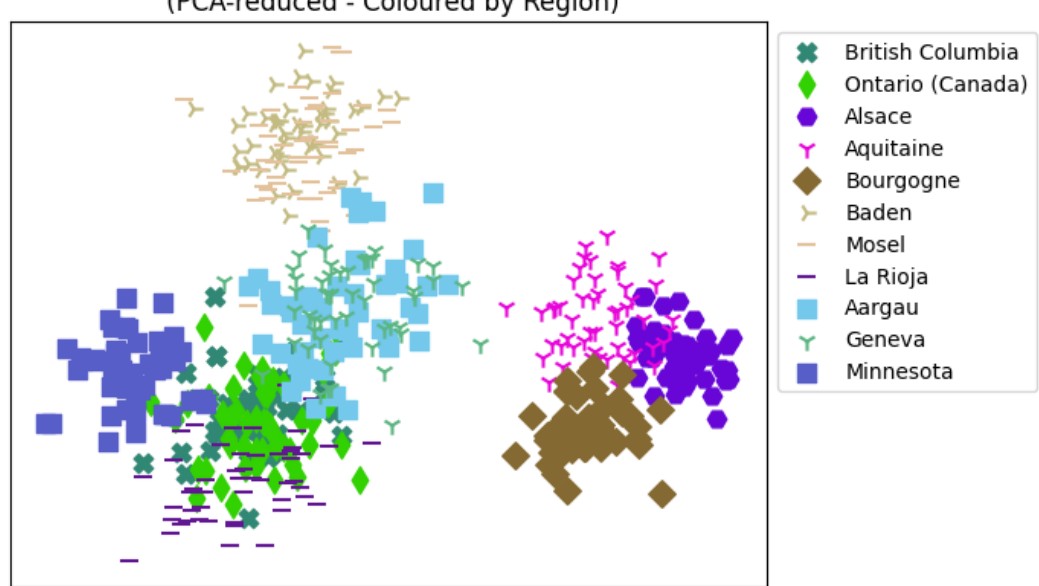

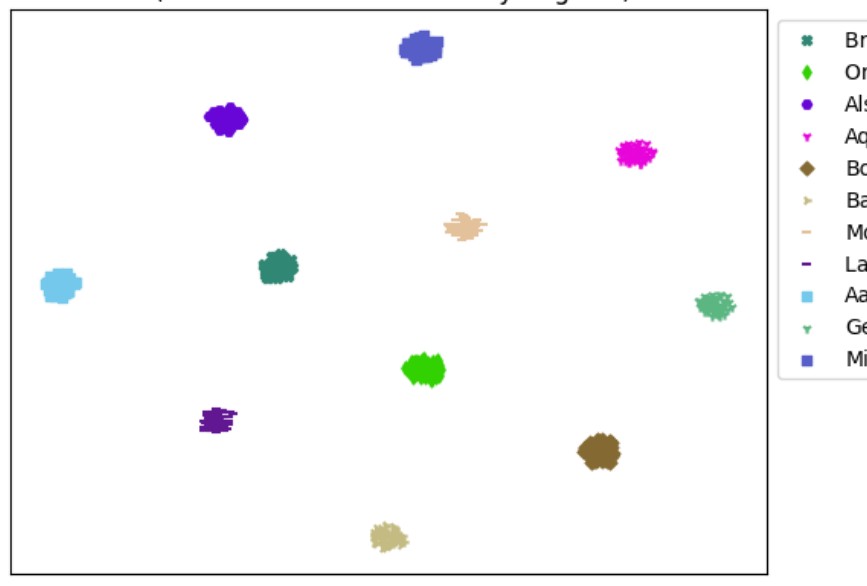

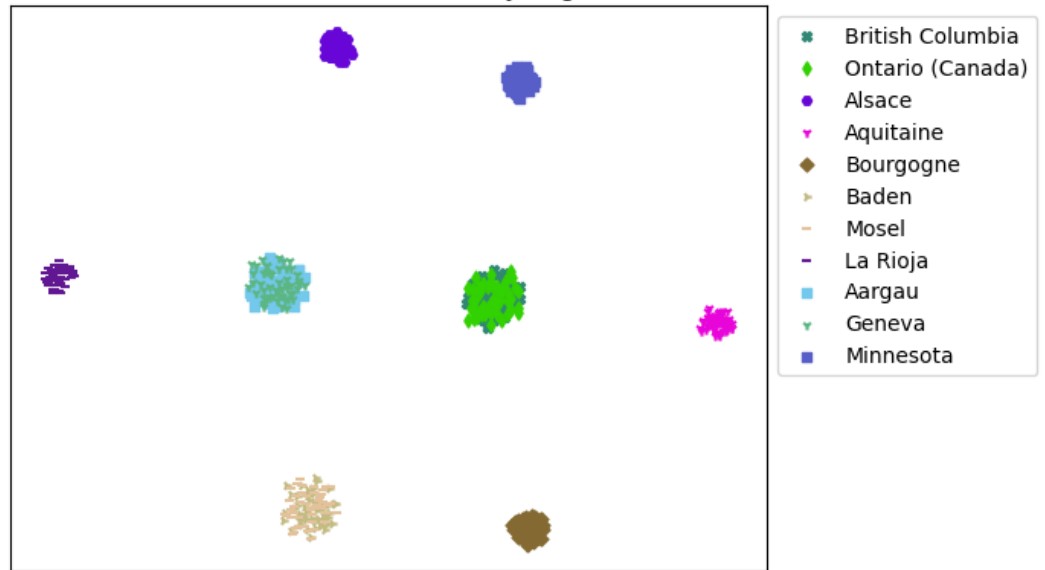

Some Regions in the Latent-Space of r_s12
(PCA-reduced - Coloured by Country)

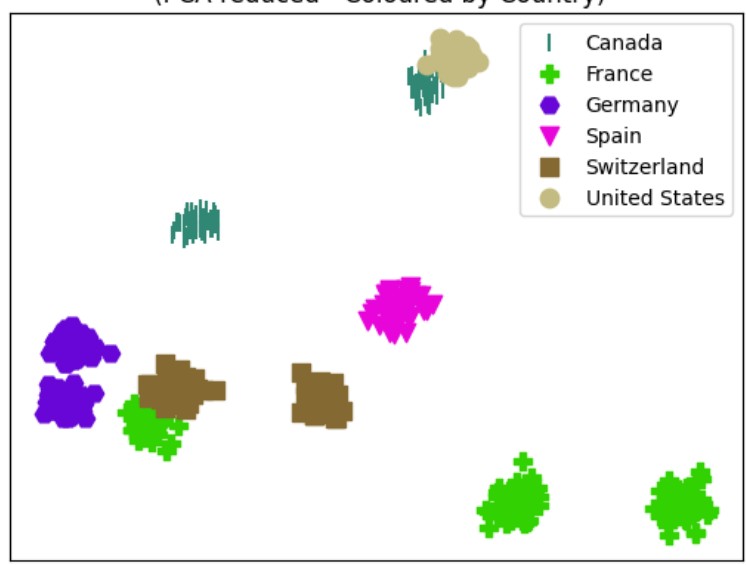

Some Regions in the Latent-Space of r_d12
(PCA-reduced - Coloured by Country)

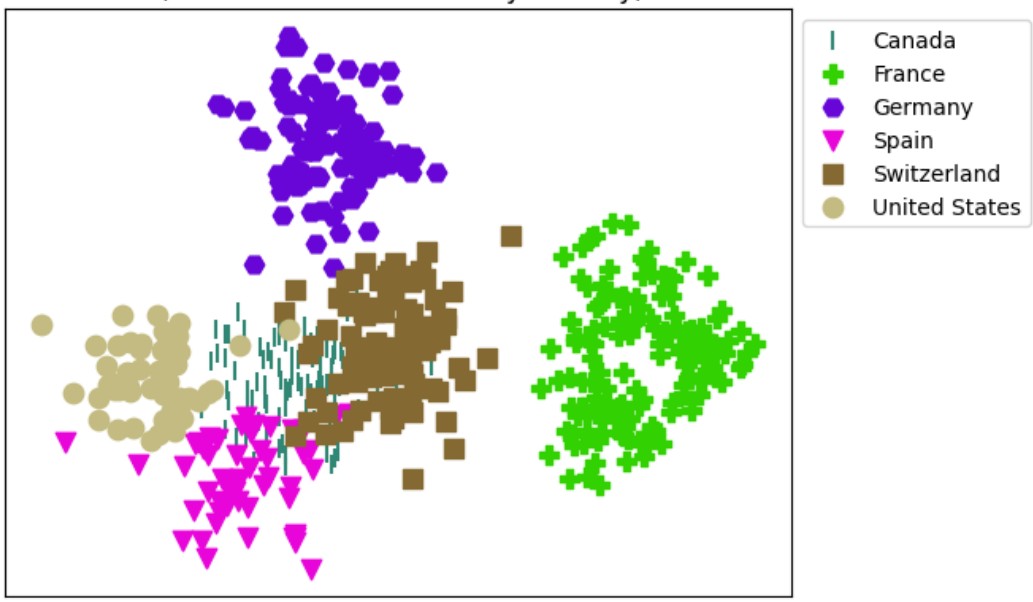

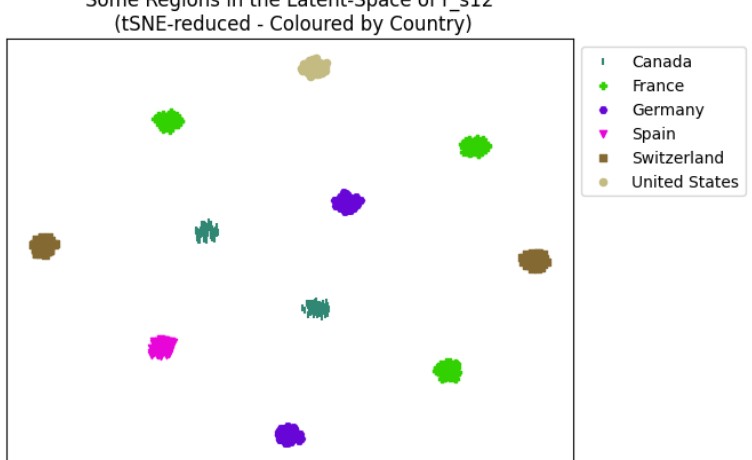

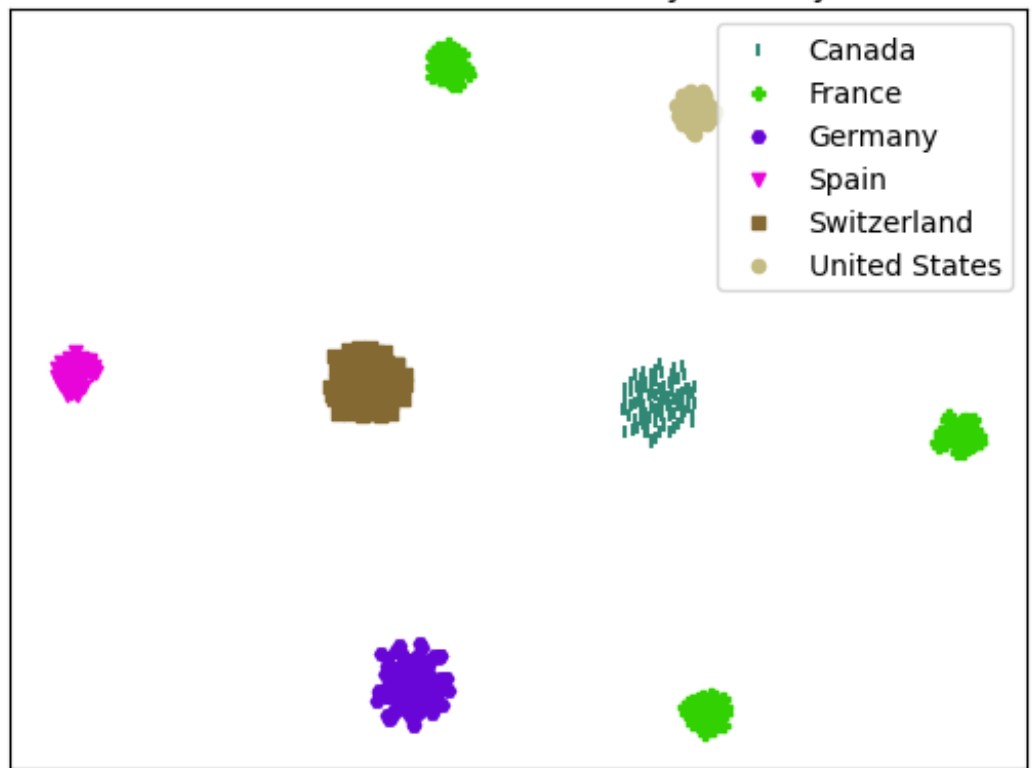

