# OpenReview forum: "Towards Deep Viticultural Representations: Joint Region and Grape Variety Embeddings"
_ICLR.cc/2024/Conference — Submitted to ICLR 2024_

### Official Review · Reviewer_p5Cm · 2023-10-26

**Soundness:** 2 fair
**Presentation:** 1 poor
**Contribution:** 1 poor
**Rating:** 3
**Confidence:** 4

**Summary:**

In the paper "Towards deep viticultural representations: joint region and grape variety embeddings", the authors develop a VAE-based approach for learning representations of wine varieties and wine-growing regions. They use data on ~600 wine-growing regions with information on which of the ~1500 wine varieties they grow. The authors train two coupled multinomial VAEs, one encoding wine variety and another encoding wine-growing region. Then the authors argue that the resulting latent representations are meaningful and outperform existing alternatives in terms of representation quality.

**Strengths:**

I am a big fan of both, representation learning and wine, so I was very excited to read this paper. I am not familiar with prior literature on wine representation learning, but the authors say it is scarce, i.e. this paper is exploring a novel application area.

**Weaknesses:**

That said, I found the paper disappointing and not really publication ready. In particular, it seems to me to fall far below the threshold of importance, novelty, and rigour expected by ICLR. The main issues are: (1) lack of clarity in writing and presentation; (2) unclear motivation for a coupled region-variety representation; (3) insufficient and unconvincing evaluation; (4) insufficient benchmark comparisons, in particular using simpler non-deep-learning baseline models.

**Questions:**

MAJOR ISSUES

1. I found the presentation very confusing. After reading the first THREE PAGES ("Introduction"), I still did not know what the authors want to achieve. I would recommend to have a clear 1-page introduction that explains what the paper wants to do (what exactly is the input data, what is the output, what are the performance metrics). The rest can go into the "Related work".

   The whole sampling approach and sampling-based evaluation are not very clear either... Does each variety (and each region) in the end get one embedding vector? Or does it get a whole distribution? Can this distribution be multimodal? All figures that I see in the Appendix contain multiple points per variety/region, but all very unimodal. Is this expected/desired?

   Many technical details are unclear as well. E.g. what K is used for K-means clustering, and why? What are the A/B/C classes used for classification? Etc.

2. The premise of the paper is that they use *coupled* VAEs for joint representations of varieties and regions. This was not motivated clearly enough. Imagine there is one wine region that grows two very different varieties. What is the perfect joint representation of this data? The "coupling" only ensures that the entire latent spaces overlap, if I understand it correctly. What would make them overlap meaningfully? I don't understand the setup or the goals here.

3. Regarding evaluation. Tables 1 and 2 assess how clustered the representations are but tell us nothing about how meaningful they are. Table 3 shows that VAEs outperform an extremely simple model based only on 2 features (latitude + longitude). And they don't outperform it very strongly. Table 4 only uses dummy baseline, Table 5 shows that all wine/region parirings are not statistically significant. So the ONLY result that actually shows that the emebddings are meaningful is Table 3; and that result is reather weak.

   I understand that quantitative evalutions may be difficult. But qualitative evaluations are lacking altogether. I would expect to see some visualisations of the latent space, but they are only shown in the Appendix and only using PCA. Moreover, they seem to make no sense! For example, in Fiture 1 in the Appendix, Alvarinho (white grape) is located close to Tempranillo (red grape), and similarly Riesling (white) is located close to Pinot Noir (red). How is this meaningful? Maybe PCA is misleading here, so why not use something like t-SNE? I feel like qualitative evaluation has not even started here.

4. Lack of proper baselines is a big problem. In Table 3, the comparison is to lon/lat model with 2 features. Why not using something that actually uses the A_ij matrix, but without deep learning? Some simple regression/classification models or maybe SVD/NNMF models applied to A_ij matrix? Same for Table 4. The authors use a relatively complex setup (coupled VAEs) for a relatively simple dataset (co-occurence matrix A_ij), so I would expect them to choose some reasonable baselines.

---

> ### Author Response · Authors · 2023-11-23
> **Response to review**
>
> Thank you for your review! We are responding to each of your comments below:
>
> 1.   	I found the presentation very confusing. After reading the first THREE PAGES ("Introduction"), I still did not know what the authors want to achieve. I would recommend to have a clear 1-page introduction that explains what the paper wants to do (what exactly is the input data, what is the output, what are the performance metrics). The rest can go into the "Related work".
> a.    *Thank you for pointing this out; we have clarified the aim of this paper and reduced and reworked the abstract and introduction sections to make them much more focused and clear.*
>
> 2.   	The whole sampling approach and sampling-based evaluation are not very clear either... Does each variety (and each region) in the end get one embedding vector? Or does it get a whole distribution? Can this distribution be multimodal? All figures that I see in the Appendix contain multiple points per variety/region, but all very unimodal. Is this expected/desired?
>
> *Each variety and region gets a multi-normal distribution that represents it. Varieties and regions share the space in which they are projected, regions and varieties that frequently occur together are pushed together by the joining losses so their distribution should overlap in the latent space. We do expect the distributions to be quite uni-modal in the dimensionality-reduced latent space, particularly when comparing “far apart” varieties or regions.*
>
> 3.   	Many technical details are unclear as well. E.g. what K is used for K-means clustering, and why? What are the A/B/C classes used for classification? Etc.
> *The A/B/C classes used for classification come from the EU legislation that determines which practices are lawful in certain regions and which aren’t. They use primarily climate data to assess this and therefore we make the claim that they implicitly contain terroir information (both through legislation and since they are based on climate and elevation features). We have clarified and expanded the explanation as to why we use this as the test set in the paper. Please see the discussion section 3.1, paragraph 1. For the K we set it to the number of labels, such that it corresponds to the number of countries, the number of varieties or the number of regions, whichever we are clustering to. We have clarified this in the figure labels and as a footnote (this section is now in the appendix, we explain more below).*
>
> 4.   	The premise of the paper is that they use coupled VAEs for joint representations of varieties and regions. This was not motivated clearly enough. Imagine there is one wine region that grows two very different varieties. What is the perfect joint representation of this data? The "coupling" only ensures that the entire latent spaces overlap, if I understand it correctly. What would make them overlap meaningfully? I don't understand the setup or the goals here.
>
> *We have gone throughout the paper to clarify the motivation, as this was pointed out by other reviewers as well. A good representation is one that relates the conditions in which grapes are grown, or the conditions under which a variety thrives to the variety itself. This way we can then use it to recommend varieties for certain growing conditions, or to more meaningfully classify varieties. Overlap is expected between varieties as we assume that in most regions multiple varieties will thrive; we do not know about all of the possible varieties that could thrive but the varieties currently grown there we assume to indicate the kind-of-varieties that do. By our definition of similarity one region would not grow two very different varieties, as the fact that they are grown near each other indicates that they thrive in a similar environment. Here for us the wine-style is not the most important indicator but the environmental conditions that a grape variety may like. This also allows clustering or classification of regions not by their historic wine style but by the environmental conditions they offer, now shown in Table 1. We also want the embeddings to be variational as this accounts for the range of environments that a grape may be found or that a region may provide. The joining of latent spaces is important as it allows imputation, or recommendations to be produced.*

---

> > ### Author Response · Authors · 2023-11-23
> > **Response to review part 2**
> >
> > 5.   	Regarding evaluation. Tables 1 and 2 assess how clustered the representations are but tell us nothing about how meaningful they are. Table 3 shows that VAEs outperform an extremely simple model based only on 2 features (latitude + longitude). And they don't outperform it very strongly. Table 4 only uses a dummy baseline, Table 5 shows that all wine/region pairings are not statistically significant. So the ONLY result that actually shows that the embeddings are meaningful is Table 3; and that result is rather weak.
> >
> > *We have adapted the SVD baseline you propose below in the new version of table 3 (table 1 now), as well as using a PCA decomposition of climate averages used in the only comparable study. As you can see we outperform these baseline much more clearly than even the previously used ones. The meaningfulness of Table 4 (now table 2) we have improved by using a new baseline based on SVD and the co-occurrence matrix as well. For Table 5 (now table 3), instead of the correlation we show the recall score in classifying varieties already grown in these regions as fitting varieties which is a more meaningful metric, in addition to the Spearman correlation significance. The non-important tables and their associated discussion we have moved to the appendix in favor of the new Qualitative Analysis section.*
> >
> > 6.   	I understand that quantitative evaluations may be difficult. But qualitative evaluations are lacking altogether. I would expect to see some visualizations of the latent space, but they are only shown in the Appendix and only using PCA. Moreover, they seem to make no sense! For example, in Figure 1 in the Appendix, Alvarinho (white grape) is located close to Tempranillo (red grape), and similarly Riesling (white) is located close to Pinot Noir (red). How is this meaningful? Maybe PCA is misleading here, so why not use something like t-SNE? I feel like qualitative evaluation has not even started here.
> >
> > *The fallacies that you point out between these grape varieties actually demonstrate that we managed to meet the aim that we set for ourselves. The reason that Alvarinho and Tempranillo should be close is because they are both very popular varieties on the Iberian peninsula, so they like warm climates. On the other hand Riesling and Pinot Noir are popularly grown near each other in the coldest of wine climates which are Germany or Canada, for example. As explained above the wine style is not the most important here but the ability to recognize grape varieties that have similar needs/potential for certain environmental conditions (though this was not clear from the initial paper). We have clarified this throughout the paper as well, and in our newly included qualitative evaluation section (section 3.3.) we discuss the relative importance of grape-color compared to the growing conditions as our model does differentiate between colors just that it first differentiates by environmental conditions and then by color.
> > Our new qualitative evaluation section is done using PCA, as we usually find it easier to interpret compared to tSNE for the similarity based models. Though in the appendix we have a comparison of PCA and tSNE reduced versions for comparison now. We sometimes find tSNE is “too good” at separating our clusters, which is why we didn’t initially use it. However, we also show/discuss the difference between PCA and tSNE views of the latent space in the Qualitative Analysis section.*
> >
> > 7.   	Lack of proper baselines is a big problem. In Table 3, the comparison is to the lon/lat model with 2 features. Why not use something that actually uses the A_ij matrix, but without deep learning? Some simple regression/classification models or maybe SVD/NNMF models applied to A_ij matrix? Same for Table 4. The authors use a relatively complex setup (coupled VAEs) for a relatively simple dataset (co-occurrence matrix A_ij), so I would expect them to choose some reasonable baselines.
> >
> > *As mentioned above we have implemented SVD baselines and shows this in our Table 3 and 4 (now Table 1 and 2). With our better performance even more significantly indicated than previous baselines. For Table 3 (now Table 1) we further incorporate a baseline based on PCA reduction of climate indices which is what the only comparable study to ours used, we again clearly outperform this baseline.*

---

> > > ### Comment · Reviewer_p5Cm · 2023-11-23
> > >
> > > I appreciate the responses. I don't have time right now to go over everything carefully (and the discussion period ends in a few hours), and I can raise my score from 1 to 3, but there seemed to be a consensus among reviewers that the paper falls short of the ICLR expectations (all four scores <=3). Hopefully the feedback was useful for future revisions and resubmissions.

---

### Official Review · Reviewer_VuWc · 2023-10-30

**Soundness:** 3 good
**Presentation:** 3 good
**Contribution:** 1 poor
**Rating:** 3
**Confidence:** 5

**Summary:**

The paper conducts a study on producing representations for viticulture (the cultivation of grapevines). It uses a VAE approach with self-supervision. Specifically, it focuses on the creation of deep embeddings for viticultural regions and grape varieties. It created tons of features for grapes.

**Strengths:**

- Interesting application to viticulture

**Weaknesses:**

- Novelty is limited on the methodological side since it aims to apply an existing method to tackle an application problem.
- No alternative baseline is used; it focuses on feature ablations of the proposed method

**Questions:**

- Can the authors formulate a machine learning challenge of this problem that is specialized for viticulture?
- Can the authors compare with other models?
- Can the authors describe novel insights about viticulture that can be gained?

---

> ### Author Response · Authors · 2023-11-23
> **Response to review**
>
> Thank you for your review! Please see our answers to your questions below:
>
> 1.   	Can the authors formulate a machine learning challenge of this problem that is specialized for viticulture?
> a.       *In the revision of this paper we have clarified the challenges that we aimed to overcome and gave a more concise and clear explanation of the state of this problem in viticulture currently. See this in the Introduction section 1.1, and the Abstract.*
> 2.   	Can the authors compare with other models?
> a.       *Yes, we now compare with a baseline using roughly the same inputs as our model that uses an SVD approach as suggested by reviewer 4, and a global region classification based on climate data and PCA analysis from the only comparable study to our knowledge. We outperform both significantly on region classification and 6/9 models outperform SVD for variety classification. Please see section 3.1 and 3.2 for this and Tables 1 and 2 for the results.*
> 3.   	Can the authors describe novel insights about viticulture that can be gained?
> a.     *In our paper, though we do not necessarily uncover surprising new results as this is a bit outside of the current scope, however, we show some latent space exploration and that it aligns closely with the reality of grape-growing. In this way our results indicate that the latent-space has a “real foundation” making it useful to find novel insights. See the Qualitative analysis section 3.3.*

---

### Official Review · Reviewer_VBQe · 2023-10-31

**Soundness:** 2 fair
**Presentation:** 2 fair
**Contribution:** 2 fair
**Rating:** 3
**Confidence:** 3

**Summary:**

This paper presents a self-supervised approach to learning joint regional and varietal embeddings using joint variational autoencoder (VAE) networks, and examines the embeddings, their usability for downstream tasks as well as whether the joint autoencoder network may be used as a varietal suitability ranking system. The results demonstrate that the embeddings to outperform ’raw’ features on downstream tasks and results indicating potential of the autoencoder networks as data-based recommender systems.

**Strengths:**

This paper applies joint variational autoencoder (VAE) networks to the study of viticultural regions and grape variety, demonstrating that the embeddings in the paper outperform the "raw" features on downstream tasks. This paper examine whether the joint autoencoder network may be used as a varietal suitability ranking system.

**Weaknesses:**

1. The English of the manuscript must be improved. There are problems with context transition and logical cohesion, and there are errors in the use of proper nouns. The methods and framework employed in the paper exhibit limited originality and innovation.
2. Section 1.1 extensively discusses the current research status in the field of viticulture, which is less relevant to the research of this paper.
3. Section 1.2 introduces the development process of embedding in detail, lacking an introduction to related technologies and methods.
4. The section 1.3 on "REGION AND VARIETY EMBEDDING REQUIREMENTS" is excessively lengthy and lacks emphasis on key points. It occupies an entire page, which is excessive.
5. There are also some unclear and unreasonable statements in the article:
1) The formulas used in this paper are not numbered.
2) There are errors in the description of variable v_i.
3) The description of the variables in the formulas is unclear.
4) Confusing organization of content in the part of the paper that tests the model on downstream tasks.
6. The description of specific parameter settings in section 2.3 is excessively lengthy. It is recommended to provide these details in APPENDIX.
7. The authors should compare their methodology with existing approaches, both qualitatively and quantitatively, to demonstrate its advantages or innovations in the field.
8. The contribution, the authors claim, to overcome the lack of detailed data. But, quantitative comparison of the above argument is hard to find in the discussion about the effectiveness in the experiments. The authors are strongly suggested to show the strength of the article in overcoming the lack of detailed data. In addition, time complexity or computation overhead analyses need be discussed properly.

**Questions:**

Please refer to the weakness section.

---

> ### Author Response · Authors · 2023-11-23
> **Response to review**
>
> Thank you for your comments! We are replying to each in detail below:
>
> 1.   	The English of the manuscript must be improved. There are problems with context transition and logical cohesion, and there are errors in the use of proper nouns.
> a.       *We have, during the current revision, worked to improve this and are going over the spelling and grammar until the final “printer-ready” submission deadline as well. Thank you for pointing this out.*
> 2.   	The methods and framework employed in the paper exhibit limited originality and innovation.
> a.       *In a way these methods are not strictly novel, however, applying joint variational autoencoders for purely categorical data is novel and requires adaptations to this domain that we describe, such as balancing the reconstruction and the KL-divergence loss terms. Furthermore, we utilize tree-structured Parzen estimator optimization for balancing the joining loss terms whereas the current literature uses purely hand-picked values to balance losses. See for example these papers: https://doi.org/10.1038/s41467-023-38125-0 and https://doi.org/10.1038/s41467-023-39895-3 . These innovations have previously not been well highlighted in our paper but we have adjusted that and emphasized this throughout, especially in the loss section of the methodology.*
> 3.   	Section 1.1 extensively discusses the current research status in the field of viticulture, which is less relevant to the research of this paper.
> a.    *We have largely removed this information and replaced it with more relevant information regarding the problem we are seeking to address. We further combined Section 1.1 and Section 1.3 into a new section called “Motivation”. It is shorter and contains only the essential points of the previously too lengthy and unspecific sections.*
> 4.   	Section 1.2 introduces the development process of embedding in detail, lacking an introduction to related technologies and methods.
> a.       *We have adjusted this section accordingly and now focus more on the methodology rather than “general concept” of the approaches we mention.*
> 5.   	The section 1.3 on "REGION AND VARIETY EMBEDDING REQUIREMENTS" is excessively lengthy and lacks emphasis on key points. It occupies an entire page, which is excessive.
> a.       *This section has been removed and key points have been instead mentioned in the “Motivation” section.*
> 6.   	The formulas used in this paper are not numbered.
> a.       *Formulas are now numbered. Thank you for pointing this out.*
> 7.   	There are errors in the description of variable v_i. The description of the variables in the formulas is unclear.
> a.       *We are a bit unsure what you mean regarding the variable descriptions, we have gone over them. Please let us know whether it is still unclear.*
> 8.   	Confusing organization of content in the part of the paper that tests the model on downstream tasks.
> a.       *We have done a major rewrite of the results and discussion section as we have received multiple comments about this. We focused particularly on clarifying and justifying the criteria and choice of tasks as well as organizing them more clearly and removing tests that are not very valuable while including a new Qualitative Analysis section.*
> 9.   	The description of specific parameter settings in section 2.3 is excessively lengthy. It is recommended to provide these details in APPENDIX.
> a.       *Other reviewers asked for more information on how these parameters are found so we have provided that in this revision, but we moved the re-formatted table listing the values to the appendix to not take up too much space.*

---

> > ### Author Response · Authors · 2023-11-23
> > **Response to review part 2**
> >
> > 10.   The authors should compare their methodology with existing approaches, both qualitatively and quantitatively, to demonstrate its advantages or innovations in the field.
> >
> > a.       *We have changed the baseline that we are using to a classical approach (i.e., SVD and PCA on climate data for the tests) more similar to the one currently used in the field and are outperforming them (Table 1 and 2 and section 3.1 and 3.2). Additionally, we have added a qualitative evaluation section (section 3.3) examining the reduced latent-spaces of selected models. Lastly, we establish a new baseline for a data-based variety recommender system, which is the first of its kind, though it leaves room for improvement (section 3.4).*
> >
> > 11.    The contribution, the authors claim, to overcome the lack of detailed data. But, quantitative comparison of the above argument is hard to find in the discussion about the effectiveness in the experiments. The authors are strongly suggested to show the strength of the article in overcoming the lack of detailed data. In addition, time complexity or computation overhead analyses need be discussed properly.
> >
> > a.     *The lack of detailed data is partially overcome by our methods as can be seen by it out-performing the climate-based PCA on climate-based region classification (see Table 1 and section 3.1). Furthermore, we explain that using a similar sampling approach used here being also applied to other modalities and terroir characteristics we can further improve our result, not by knowing how many grapes grow on which soil but by knowing which regions have primary which varieties and primarily which soil and therefore match grape varieties to soil indirectly. Which is why this approach is so important, as it demonstrates the underlying methodology for an expandable terroir embedding and grape recommendation system.*
> >
> > b. *Time complexity is comparable to previously published research on joint VAEs, depending mainly on the loss-function calculation. We have now pointed this out in the methodology (section 2.3). For the computation requirements we included the training time and hardware in a table in the appendix for each model. Unfortunately we did not have the space to also go into more detail about this in the discussion section.*

---

### Official Review · Reviewer_U2kf · 2023-11-01

**Soundness:** 2 fair
**Presentation:** 2 fair
**Contribution:** 2 fair
**Rating:** 3
**Confidence:** 4

**Summary:**

The paper proposes to jointly learn continuous representations for viticultural regions and grape varieties, which can potentially be leveraged in downstream tasks for enhanced performance. Specifically, the paper first constructs a grape variety and region co-occurrence dataset, and then uses a variational auto-encoder (VAE) model to learn the low-dimensional representations. The model is trained using the VAE loss and a joining loss.

**Strengths:**

1. The idea of learning low-dimensional representations for viticultural regions and grape varieties to improve the performance of downstream tasks seems novel and interesting.
2. The paper also conducted experiments to investigate the property of the latent representation space.

**Weaknesses:**

1. The writing of the paper could be improved as it is sometimes difficult to follow the paper.
2. The paper is incremental since it simply applies VAE to learn the latent representations of grape varieties and viticultural regions.
3. I think the joining loss is quite important in aligning the representations of regions and varieties. The paper introduces three different joining losses. However, the details of these losses are missing from the paper.
4. The paper lists the weights of different losses in the paper. However, it is unclear how these weights are chosen.
5. I find the Results & Discussion section difficult to follow. There is no explicit introduction to the datasets, baselines, experimental setups and research questions.

**Questions:**

Please see the questions in the Weaknesses section.

---

> ### Author Response · Authors · 2023-11-23
> **Response to review**
>
> Thank you for your review. Please see our responses to your comments below:
>
> 1.   	The writing of the paper could be improved as it is sometimes difficult to follow the paper.
> a.       *Thank you for pointing this out, we are revising the grammar of the paper and have also re-written the abstract to be clearer. We are double checking throughout while implementing the other changes.*
>
> 2.   	The paper is incremental since it simply applies VAE to learn the latent representations of grape varieties and viticultural regions.
> a.       *We introduce a joint VAE approach for both learning latent representations to assist with classification of viticultural regions (where the only precedent was PCA decomposition) and of grape varieties for which, to our knowledge, no such classification that is based on utilization of the grapes (rather than the origin of them) exists. Lastly, we present a methodology that shows promise of making data-based recommendations to inform varietal choice for vineyard developers under climate uncertainty, this is completely novel. We have clarified aims and innovations in the revised abstract, and in the rest of the paper as well (see motivation and conclusion for example). Additionally, we show the application of joint VAEs to purely categorical data which, to our knowledge, has also not been done before; this required us to re-balance the VAE loss prior to being able to determine the joining loss weights. The joining weights we find by Bayesian optimization rather than hand-picking as is the norm in the current literature such as: https://doi.org/10.1038/s41467-023-38125-0  and https://doi.org/10.1038/s41467-023-39895-3 . This we have highlighted as well (see section 2.3, last paragraph).*
>
> 3.   	I think the joining loss is quite important in aligning the representations of regions and varieties. The paper introduces three different joining losses. However, the details of these losses are missing from the paper.
> a.     *We have gone over the loss-section of the methodology and provide some additional details about the similarity and correlation losses. Please let us know whether this is now more appropriate.*
>
> 4.   	The paper lists the weights of different losses in the paper. However, it is unclear how these weights are chosen.
> a.       *We find the joining loss weights by tree-structured Parzen estimator optimization. We have made this more clear and explicit in the paper, and included our optimization criterion. Please see section 2.3*
>
> 5.   	I find the Results & Discussion section difficult to follow. There is no explicit introduction to the datasets, baselines, experimental setups and research questions.
> a.      *The discussion and results were revised according to your and other reviewer comments. See section 3.1, 3.2 and 3.4 for the experiments and baselines.*

---

### Author Response · Authors · 2023-11-23
**Major Revision**

Dear Reviewers,

Thank you for your comments. We have put the paper through major revision to adjust it according to your comments and to address the major shortcomings. The new version is re-focused, simplified by removing unnecessary information and solidified by inclusion of proper baselines as well as a qualitative analysis section.

We are also replying to each of the reviewers individually to address their specific concerns.

---

### Meta-Review · Area_Chair_kvzr · 2023-12-05

**Metareview:**

The paper is interested in proposing a comprehensive description of viticultural characteristics, based on a VAE. According to all reviewers, the paper "falls short of the ICLR expectations", and the reviewers and area chair hope that the reviews will help the authors to better situate their contributions and present a comparative assessment thereof in a further version of the paper.
Advice: in the rebuttal, it is often more effective to answer in a detailed way to the comments/questions, rather than saying "we made modifications in the paper to address this comment".

**Justification For Why Not Higher Score:**

--

**Justification For Why Not Lower Score:**

N/A

---

### Decision · Program_Chairs · 2024-01-16

Reject